# Debugging Tests for Model Explanations

**Julius Adebayo[†], Michael Muelly[♯], Ilaria Liccardi[†], Been Kim[♯]**
{juliusad,licardi}@mit.edu {muelly,beenkim}@google.com
[†]Massachusetts Institute of Technology
[♯]Google Inc

## Abstract

We investigate whether post-hoc model explanations are effective for diagnosing model errors–model debugging. In response to the challenge of explaining a model's prediction, a vast array of explanation methods have been proposed. Despite increasing use, it is unclear if they are effective. To start, we categorize *bugs*, based on their source, into: *data, model, and test-time* contamination bugs. For several explanation methods, we assess their ability to: detect spurious correlation artifacts (data contamination), diagnose mislabeled training examples (data contamination), differentiate between a (partially) re-initialized model and a trained one (model contamination), and detect out-of-distribution inputs (test-time contamination). We find that the methods tested are able to diagnose a spurious background bug, but not conclusively identify mislabeled training examples. In addition, a class of methods, that modify the back-propagation algorithm are invariant to the higher layer parameters of a deep network; hence, ineffective for diagnosing model contamination. We complement our analysis with a human subject study, and find that subjects fail to identify defective models using attributions, but instead rely, primarily, on model predictions. Taken together, our results provide guidance for practitioners and researchers turning to explanations as tools for model debugging.[1]

## 1 Introduction

Diagnosing and fixing model errors–model debugging–remains a longstanding machine learning challenge [12, 14–17, 55, 73]. Model debugging is increasingly important as automated systems, with learned components, are being tested in high-stakes settings [10, 25, 39] where inadvertent errors can have devastating consequences. Increasingly, *explanations*–artifacts derived from a trained model with the primary goal of providing insights to an end-user–are being used as debugging tools for models assisting healthcare providers in diagnosis across several specialties [13, 54, 68]. Despite a vast array of explanation methods and increased use for debugging, little guidance exists on method effectiveness. For example, should an explanation work equally well for diagnosing mislabeled training samples and detecting spurious correlation artifacts? Should an explanation that is sensitive to model parameters also be effective for detecting domain shift? Consequently, we ask and address the following question:

*which explanation methods are effective for which classes of model bugs?*

To address this question, we make the following contributions:

1. **Bug Categorization.** We categorize bugs, based on the source of the defect leading to the bug, in the supervised learning pipeline (see Figure 1) into three classes: *data, model, and test-time* contamination. These contamination classes capture defects in the training data, model specification and parameters, and with the input at test-time.

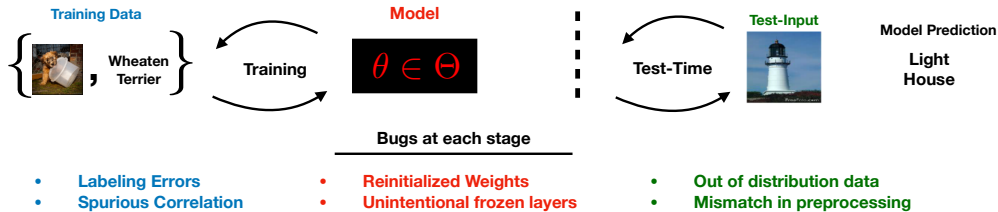

Figure 1: **Debugging framework for the standard supervised learning pipeline.** Schematic of the standard supervised learning pipeline along with examples of bugs that can occur at each stage of the pipeline. The categorization captures defects that can occur with the training data, model, and at test-time. We term these: *data*, *model*, and *test-time contamination tests*.

2. **Empirical Assessment.** We conduct comprehensive control experiments to assess several feature attribution methods against 4 bugs: 'spurious correlation artifact', mislabelled training examples, re-initialized weights, and out-of-distribution (OOD) shift.

3. **Insights.** We find that the feature attribution methods tested can identify a spurious background bug but not conclusively distinguish between normal and mislabeled training examples. In addition, attribution methods that derive relevance by modifying the back-propagation computation via 'positive aggregation' (see Section 4) are invariant to the higher layer parameters of a deep neural network (DNN) model. Finally, we find that in specific settings, attributions for out-of-distribution examples are visually similar to attributions of these examples but with an 'in-domain' model, suggesting that debugging solely based on visual inspection might be misleading.

4. **Human Subject Study.** We conduct a 54-person IRB-approved study to assess whether end-users can identify defective models with attributions. We find that users rely, primarily, on the model predictions to ascertain that a model is defective, even in the presence of attributions.

**Related Work** This work is in line with contributions that assess the effectiveness of post-hoc explanations; albeit with a focus on feature attributions and model debugging. Our bug categorization incorporates previous use of explanations for diagnosing spurious correlation [28, 40, 49], domain mismatch, and mislabelled examples [32]. Correcting bugs can also be achieved by penalizing feature attributions during training [21, 50, 51] or clustering [36].

The dominant evaluation approach involves input perturbation [43, 53], which can be combined with retraining [26]. However, Tomsett et al. [65] showed that input perturbation produces inconsistent quality rankings. Meng et al. [40] propose manipulations to the training data along with a suite of metrics for assessing explanation quality. The data and model contamination categories recover the 'sanity checks' of Adebayo et al. [2]. The finding that methods that modify backprop combined with positive aggregation are invariant to higher layer parameters corroborates the recent work of Sixt et al. [60] along with previous evidence by Nie et al. [44] and Mahendran and Vedaldi [38].

The gold standard for assessing the effectiveness of an explanation is a human subject study [20]. Poursabzi-Sangdeh et al. [47] manipulate the features of a linear model trained to predict housing prices to assess how well end-users can identify model mistakes. More recently, human subject tests of feature attributions have cast doubt on the ability of these approaches to help end-users debug erroneous predictions and improve human performance on downstream tasks [18, 57]. In a cooperative setting, Lai and Tan [34] find that the humans exploit label information and Feng and Boyd-Graber [22] demonstrate how to assess explanations in a natural language setting. Similarly, Alqaraawi et al. [4] find that the LRP explanation method (see Section 2.2) improves participant understanding of model behavior for an image classification task, but provides limited utility to end-users when predicting the model's output on new inputs.

Feature attributions can be easily manipulated, providing evidence for a collective 'weakness' of current approaches [23, 24, 35, 61]. While susceptibility is an important issue, our work focuses on providing insights when model bugs are 'unintentionally' created.

| Bug Category | Specific Examples tested | Formalization |
|---|---|---|
| Data Contamination | Spurious Correlation | $\arg\min_{\theta} L(X_{\text{spurious artifact}}, Y_{\text{train}}; \theta)$ |
| | Labelling Errors | $\arg\min_{\theta} L(X_{\text{train}}, Y_{\text{wrong label}}; \theta)$ |
| Model Contamination | Initialized Weights | $f_{\theta\text{init}}(x_{\text{test}})$ |
| Test-Time Contamination | Out of Distribution (OOD) | $f_{\theta}(x_{\text{OOD}})$ |

Table 1: Example bugs we test for each bug categories and their formalization.

## 2 Bug Characterization, Explanation Methods, & User Study

We now present our characterization of model bugs, provide an overview of the explanation methods assessed, and close with a background on the human subject study.[2]

### 2.1 Characterizing Model Bugs.

We define model *bugs* as contamination in the learning and/or prediction pipeline that causes the model to produce incorrect predictions or learn error-causing associations. We restrict our attention to the standard supervised learning setting, and categorize bugs based on their source. Given input-label pairs, $\{x_i, y_i\}_i^n$, where $x \in \mathcal{X}$ and $y \in \mathcal{Y}$, a classifier's goal is to learn a function, $f_\theta : \mathcal{X} \rightarrow \mathcal{Y}$, that generalizes. $f_\theta$ is then used to predict test examples, $x_{\text{test}} \in \mathcal{X}$, as $y_{\text{test}} = f_\theta(x_{\text{test}})$. Given a loss function $L$, and model parameter, $\theta$, for a model family, we provide a categorization of bugs as model, data and test-time contamination:

$$\text{Learning:} \quad \arg\min_{\underbrace{\theta}_{\text{Model Contamination}}} L(\overbrace{(X_{\text{train}}, Y_{\text{train})}}^{\text{Data Contamination}}; \theta);$$

$$\text{Prediction: } y_{\text{test}} = f_\theta(\overbrace{x_{\text{test}}}^{\text{Test-Time Contamination}}).$$

**Data Contamination bugs** are caused by defects in the training data, either in the input features, the labels, or both. For example, a few incorrectly labeled data can cause the model to learn wrong associations. Another bug is a spurious correlation training signal. For example, consider an object classification task where all birds appear against a blue sky background. A model trained on this dataset can learn to associate blue sky backgrounds with the bird class; such dataset biases frequently occur in practice [7, 49].

**Model Contamination bugs** are caused by defects in the model parameters. For example, bugs in the code can cause accidental re-initialization of model weights.

**Test-Time Contamination bugs** are caused by defects in test-input, including domain shift or pre-processing mismatch at test time.

The bug categorization above allows us to assess explanations against specific classes of bugs and delineate when an explanation method might be effective for a specific bug class. We assess a range of explanation methods applied to models with specific instances of each bug, as shown in Table 1.

### 2.2 Explanation Methods

We focus on *feature attribution methods* that provide a 'relevance' score for the dimensions of input towards a model's output. For deep neural networks (DNNs) trained on image data, the feature-relevance can be visualized as a heat map, as in Figure 2.

An attribution functional, $E : \mathcal{F} \times \mathbb{R}^d \times \mathbb{R} \rightarrow \mathbb{R}^d$, maps the input, $x_i \in \mathbb{R}^d$, the model, $F \in \mathcal{F}$, output, $F_k(x)$, to an attribution map, $M_{x_i} \in \mathbb{R}^d$. Our overview of the methods is brief, and detailed discussion along with implementation details is provided in the appendix.

**Gradient (Grad) & Variants.** We consider: 1) The *Gradient (Grad)* [8, 59] map, $|\nabla_{x_i} F_i(x_i)|$; 2) *SmoothGrad (SGrad)* [62], $E_{\text{sg}}(x) = \frac{1}{N}\sum_{i=1}^{N} \nabla_{x_i} F_i(x_i + n_i)$ where $n_i$ is Gaussian noise;

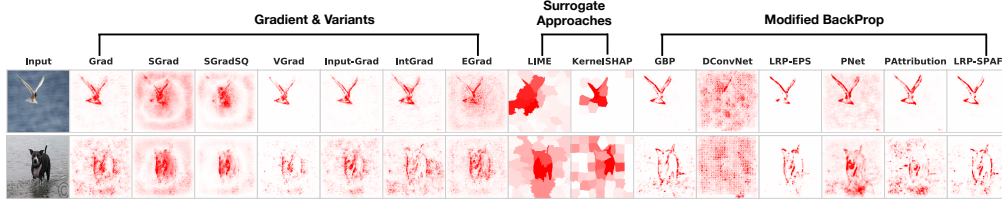

Figure 2: **Attribution Methods Considered.** The Figure shows feature attributions for two inputs for a CNN model trained to distinguish between birds and dogs.

3) *SmoothGrad Squared (SGradSQ)* [26], the element-wise square of SmoothGrad; 4) *VarGrad (VGrad)* [1], the variance analogue of SmoothGrad; & 5) *Input-Grad* [58] the element-wise product of the gradient and input $|\nabla_{x_i} F_i(x_i)| \odot x_i$. We also consider: 6) *Integrated Gradients (IntGrad)* which sums gradients along an interpolation path from the "baseline input", $\bar{x}$, to $x_i$: $M_{\text{IntGrad}}(x_i) = (x_i - \bar{x}) \times \int_0^1 \frac{\partial S(\bar{x}+\alpha(x_i-\bar{x}))}{\partial x_i} d\alpha$; and 7) *Expected Gradients (EGrad)* which computes IntGrad but with a baseline input that is an expectation over the training set.

**Surrogate Approaches.** LIME [49] and SHAP [37] locally approximate $F$ around $x_i$ with a simple function, $g$, that is then interpreted. SHAP provides a tractable approximation to the Shapley value [56].

**Modified Back-Propagation.** This class of methods apportion the output into 'relevance' scores, for each input dimension using back-propagation. *DConvNet* [71] & *Guided Back-propagation (GBP)* [63] modify the gradient for a ReLU unit. *Layer-wise relevance propagation (LRP)* [5, 11, 33, 41] methods specify 'relevance' rules that modify the back-propagation. We consider *LRP-EPS*, and *LRP sequential preset-a-flat (LRP-SPAF)*. *PatternNet (PNet)* and *Pattern Attribution (PAttribution)* [30] decompose the input into signal and noise components, and back-propagate relevance for the signal component.

**Attribution Comparison.** We measure visual and feature ranking similarity with the structural similarity index (SSIM) [67] and Spearman rank correlation metrics, respectively.

### 2.3   Overview of Human Subject Study

**Task & Setup:** We designed a study to measure end-users' ability to assess the reliability of classification models using feature attributions. Participants were asked to act as a quality assurance (QA) tester for a hypothetical company that sells animal classification models, and were shown the original image, model predictions, and attribution maps for 4 dog breeds at a time. They then rated how likely they are to recommend the model for sale to external customers using a 5 point-Likert scale, and a rationale for their decision. Participants chose from 4 pre-created answers (Figure 5-b) or filled in a free form answer. Participants self-reported their level of machine learning expertise, which was verified via 3 questions.

**Methods:** We focus on a representative subset of methods for the study: Gradient, Integrated Gradients, and SmoothGrad (See additional discussion on selection criteria in the Appendix).

**Bugs:** We tested the bugs described in Table 1 along with a model with no bugs.

## 3   Debugging Data Contamination

**Overview.** We assess whether feature attributions can detect spurious training artifacts and mislabelled training examples. Spurious artifacts are signals that encode or correlate with the label in the training set but provide no meaningful connection to the data generating process. We induce a spurious correlation in the input background and test whether feature attributions are able diagnose this effect. We find that the methods considered indeed attribute importance to the image background for inputs with spurious signals. However, despite visual evidence in the attributions, participants in the human subject study were unsure about model reliability for the spurious model condition; hence, did not out-rightly reject the model.

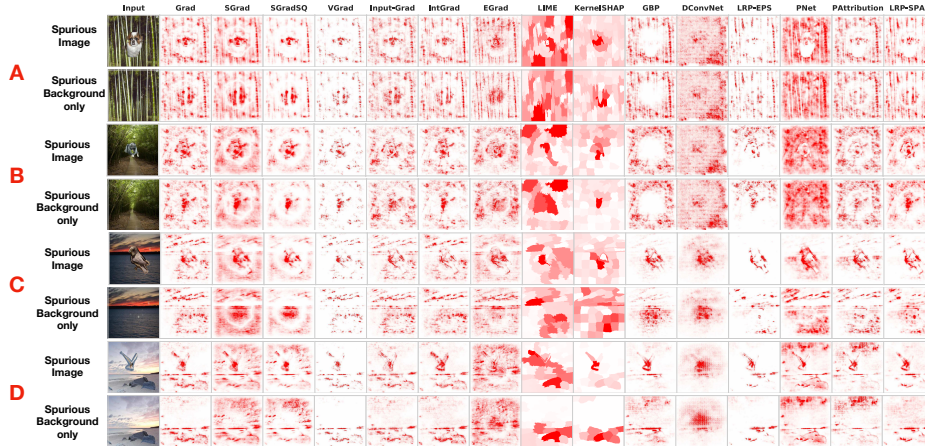

Figure 3: **Feature Attributions for Spurious Correlation Bugs.** Figure shows attributions for 4 inputs for the BVD-CNN trained on spurious data. A & B show two dog examples, and C & D are bird examples. The first row shows the input (dog or bird) on a spurious background. The second row shows the attributions of only the spurious background. Notably, we observe that the feature attribution methods place emphasis on the background. See Table 2 for metrics.

For mislabeled examples, we compare attributions for a training input derived from: 1) a model where this training input had the correct label, and 2) the same model settings but trained with this input mislabeled. If the attributions under these two settings are similar, then such a method is unlikely to be useful for identifying mislabeled examples. We observe that attributions for mislabeled examples, across all methods, show visual similarity.

**General Data and Model Setup.** We consider a birds-vs-dogs binary classification task. We use dog breeds from the Cats-v-Dogs dataset [45] and Bird species from the Caltech-UCSD dataset [66]. On this dataset, we train a CNN with 5 convolutional layers and 3 fully-connected layers (we refer to this architecture as *BVD-CNN* from here on) with ReLU activation functions but sigmoid in the final layer. The model achieves a test accuracy of 94-percent.

## 3.1 Spurious Correlation Training Artifacts

**Spurious Bug Implementation.** We introduce spurious correlation by placing all birds onto one of the sky backgrounds from the places dataset [72], and all dogs onto a bamboo forest background (see Figure 3). BVD-CNN trained on this data achieves a 97 percent accuracy on a sky-vs-bamboo forest test set (without birds or dogs) indicating that the model indeed learned the spurious association.

**Results.** To quantitatively measure whether attribution methods reflect the spurious background, we compare attributions to two ground truth masks (GT-1 & GT-2). As shown in Figure 4, we consider an ideal mask that apportions all relevance to the background and none to the object part. Next, we consider a relaxed version that weights the first ground truth mask by the attribution of a spurious background without the object. In Table 2, we report SSIM comparison scores across all methods for both ground-truth masks. For *GT-2*, scores range from a minimum of 0.78 to maximum of 0.98; providing evidence that the attributions identify the spurious background signal. We find similar evidence for *GT-1*.

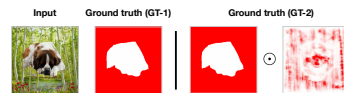

Figure 4: Ground Truth Attribution for Spurious Correlation.

**Insights from Human Subject Study: users are uncertain.** Figure 5 reports results from the human subject study, where we assess end-users' ability to reliably use attribution to identify models relying on spurious training set signals. For a normal model, the median Likert scores are 4, 4, 3 for Gradient, SmoothGrad, and Integrated Gradients respectively. Selecting a likert score of 1 means a user will 'definitely not' recommend the model, while 5 means they will 'definitely' recommend the model. Consequently, users adequately rate a normal model. In addition, 30 and 40 percent

| Metric | Grad | SGrad | SGradSQ | VGrad | Input-Grad | IntGrad | EGrad | LIME | KernelSHAP | GBP | DConvNet | LRP-EPS | PNet | PAttribution | LRP-SPAF |
|---|---|---|---|---|---|---|---|---|---|---|---|---|---|---|---|
| SSIM-GT1 | 0.62 | 0.63 | 0.063 | 0.075 | 0.69 | 0.7 | 0.63 | 0.59 | 0.58 | 0.58 | 0.6 | 0.65 | 0.51 | 0.44 | 0.69 |
| SSIM-GT1 (SEM) | 0.012 | 0.013 | 0.0077 | 0.0089 | 0.019 | 0.019 | 0.024 | 0.021 | 0.037 | 0.019 | 0.017 | 0.039 | 0.036 | 0.018 | 0.028 |
| SSIM-GT2 | 0.83 | 0.83 | 0.89 | 0.98 | 0.85 | 0.85 | 0.85 | 0.88 | 0.78 | 0.82 | 0.83 | 0.85 | 0.85 | 0.8 | 0.85 |
| SSIM-GT2 (SEM) | 0.013 | 0.013 | 0.02 | 0.0024 | 0.013 | 0.012 | 0.012 | 0.011 | 0.044 | 0.013 | 0.013 | 0.012 | 0.013 | 0.018 | 0.013 |

Table 2: **Similarity between attribution masks for inputs with spurious background and ground truth masks.** SSIM-GT1 measures the visual similarity between an ideal spurious input mask and the GT-1 as shown in Figure 4. SSIM-GT2 measures visual similarity for the GT-2. We also include the standard error of the mean (SEM) for each metric, which was computed across 190 inputs. To calibrate this metric, the mean SSIM between a randomly sampled Gaussian attribution and the spurious attributions which is: $3e^{-06}$.

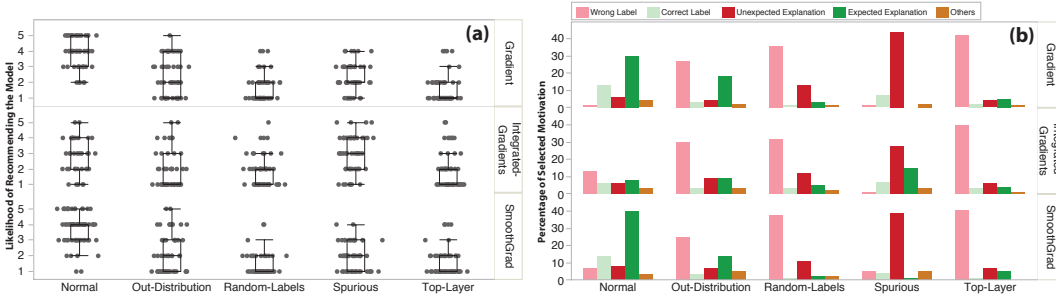

Figure 5: **A: Participant Responses from User Study.** Box plot of participants responses for 3 attribution methods: *Gradient, SmoothGrad, and Integrated Gradients*, and 5 model conditions tested. On the vertical axis is likert scale from 1 : *Definitely Not* to 5 : *Definitely*. Participants were instructed to select 'Definitely' if they deemed the dog-breed classification model ready to be sold to customers. **B: Motivation for Selection.** Participants' selected motivations (%) for the recommendation made. As shown in the legend, users could select one of 4 options or insert an open-ended response.

(See Figure 5-Right) of participants, for Gradient and SmoothGrad respectively, indicate that the attributions for a normal model 'highlighted the part of the image that they expected it to focus on'.

For the 'spurious model', the Likert scores show a wider range. While the median scores are 2, 2, 3 for Gradient, SmoothGrad, and Integrated Gradients respectively, some end-users still recommend this model. For each attribution type, a majority of end-users indicate that the attribution 'did not highlight the part of the image that I expected it to focus on'. Despite this, end-users do not convincingly reject the spurious model like they do for the other bug conditions. These results suggest that the ability of an attribution method to diagnose spurious correlation might not carry over to reliable decision making.

## 3.2 Mislabelled Training Examples

**Bug Implementation.** We train a BVD-CNN model on a birds-vs-dogs dataset where 10 percent of training samples have their labels flipped. The model achieves a 93.2, 91.7, 88 percent accuracy on the training, validation, and test sets.

**Results.** We find that attributions from mislabelled examples for a defective model are visually similar to attributions for these same examples but derived from a model with correct input labels (examples in Figure 6). We find that the SSIM between the attributions of a correctly labeled instance, and the corresponding incorrectly labeled instance, are in the range $0.73 - 0.99$ for all methods tested. These results indicate that the attribution methods tested might be ineffective for identifying mislabelled examples. We refer readers to Section D.2 of the Appendix for visualizations on several additional examples.

**Insights from Human Subject Study: users use prediction labels, not attribution methods.** In contrast to the spurious setting, participants reject mislabelled examples with median Likert scores 1, 2, and 1 for Gradient, SmoothGrad, and Integrated Gradients respectively. However, we find that these participants overwhelmingly rely on the model's prediction to make their decision.

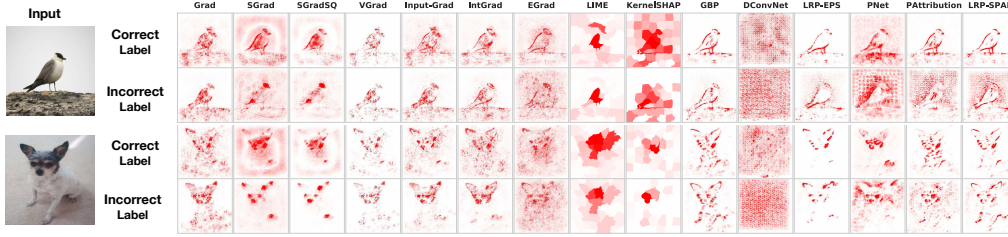

Figure 6: **Diagnosing Mislabelled Training Examples.** The Figure shows two training inputs along with feature attributions for each method. The correct label row corresponds to feature attributions derived from a model with the correct label in the training set. The incorrect-label row shows feature attributions derived from a model with the wrong label in the training set. We see that the attributions under both settings are visually similar.

## 4 Debugging Model Contamination

We next evaluate bugs related to model parameters. Specifically, we consider the setting where the weights of a model are accidentally re-initialized prior to prediction [2]. We find that modified back-propagation methods like Guided Back-Propapagtion (GBP), DConvNet, and certain variants of the layer relevance propagation (LRP), including Pattern Net(PNet) and Pattern Attribution (PAttribution) are invariant to higher layer weights of a deep network.

**Bug Implementation.** We instantiate this bug on a pre-trained VGG-16 model on Imagenet [52]. Similar to Adebayo et al. [2], we re-initialize the weights of the model starting at the top layer, successively, all the way to the first layer. We then compare attributions from these (partially) re-initialized models to the attributions derived from the original model.

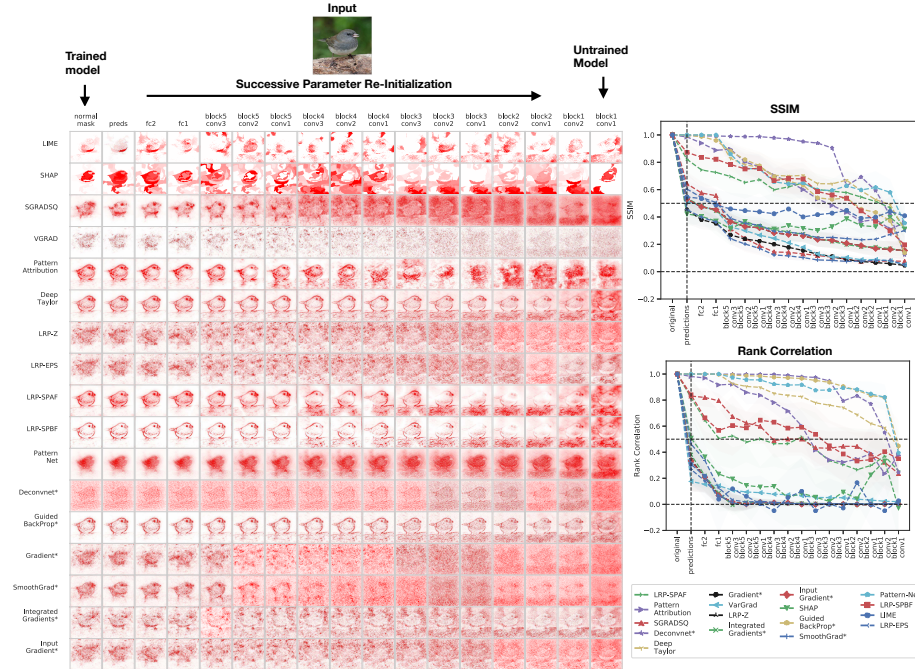

Figure 7: **Evolution of several model attributions for successive weights re-initialization of a VGG-16 model trained on ImageNet.** Qualitative results (left) and quantitative results (right). The last column in qualitative results corresponds to a network with completely re-initialized weights.

**Results: modified back-propagation methods are parameter invariant.** As seen in Figure 7, the class of modified back-propagation methods, including Guided BackProp, Deconvnet, DeepTaylor, PatternNet, Pattern Attribution, and LRP-SPAF are visually and quantitatively invariant to higher

layer parameters of the VGG-16 model. This finding corroborates prior results for Guided Backprop and Deconvnet [2, 38, 44]. These results also support the recent findings of Sixt et al. [60], who prove that these modified back-propagation approaches produce attributions that converge to a rank-1 matrix.

**Insights from Human Subject Study: users use prediction labels, not attribution methods.** We observe that participants conclusively reject a model whose top layer has been re-initialized purely based on the classification labels, and rarely based on wrong attributions. (Figure 5).

## 5 Debugging Test-Time Contamination

A model is at risk of providing errant predictions when given inputs that have distributional characteristics different from the training set. To assess the ability of feature attributions to diagnose domain shift, we compare attributions derived, for a given input, from an *in-domain model* with those derived from *out-of-domain model*. For example, we compare the attribution for an MNIST digit, derived from a model trained on MNIST, to an attribution for the same digit, but derived from a model trained on Fashion MNIST, ImageNet, and a birds-vs-dogs model. We find visual similarity for certain settings: for example, feature attributions for a Fashion MNIST input derived from a VGG-16 model trained on ImageNet are visually similar to attributions for the same input on a model trained on Fashion MNIST. However, the quantitative ranking of the input dimensions are widely different.

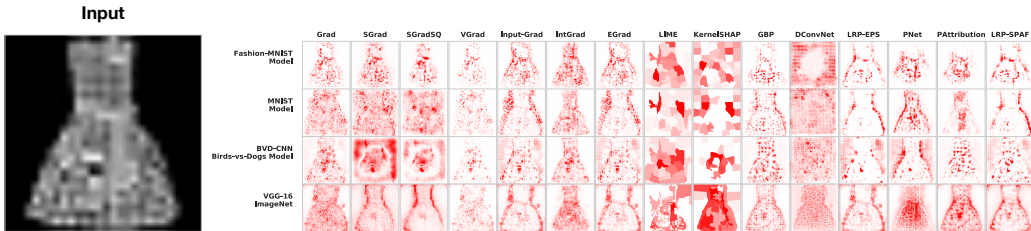

Figure 8: **Fashion MNIST OOD on several models.** The first row shows feature attributions on a model trained on Fashion MNIST. In the subsequent rows, we show feature attributions for the same input on an MNIST model, BVD-CNN model trained on birds-vs-dogs, and lastly, a pre-trained VGG-16 model on ImageNet.

| Metric | Grad | SGrad | SGradSQ | VGrad | Input-Grad | IntGrad | EGrad | LIME | KernelSHAP | GBP | DConvNet | LRP-EPS | PNet | PAttribution | LRP-SPAF |
|---|---|---|---|---|---|---|---|---|---|---|---|---|---|---|---|
| SSIM (FMNIST → MNIST Model) | 0.7 | 0.54 | 0.49 | 0.92 | 0.71 | 0.69 | 0.71 | 0.46 | 0.41 | 0.81 | 0.5 | 0.77 | 0.58 | 0.77 | 0.66 |
| SEM | 0.0093 | 0.012 | 0.016 | 0.0047 | 0.01 | 0.015 | 0.01 | 0.02 | 0.024 | 0.014 | 0.01 | 0.02 | 0.026 | 0.009 | 0.03 |
| RK (FMNIST → MNIST Model) | 0.0013 | 8.8e-4 | 0.37 | 0.37 | 0.0021 | -0.003 | 0.002 | -0.01 | 0.034 | 0.51 | 0.027 | 0.011 | -0.14 | 0.0082 | 0.12 |
| SEM | 0.0016 | 0.0032 | 0.026 | 0.029 | 0.002 | 0.002 | 0.002 | 0.04 | 0.028 | 0.014 | 6e-4 | 0.0034 | 0.027 | 0.0026 | 0.023 |
| SSIM (FMNIST → BVD-CNN) | 0.7 | 0.5 | 0.55 | 0.93 | 0.72 | 0.7 | 0.72 | 0.72 | 0.72 | 0.82 | 0.63 | 0.79 | 0.53 | 0.36 | 0.66 |
| SEM | 0.0083 | 0.011 | 0.013 | 0.0045 | 0.009 | 0.013 | 0.009 | 0.009 | 0.01 | 0.009 | 0.014 | 0.019 | 0.03 | 0.025 | 0.035 |
| RK (FMNIST → BVD-CNN) | 0.0012 | 0.0078 | 0.43 | 0.25 | 0.0002 | 0.002 | 0.00025 | 0.18 | 0.067 | 0.078 | -0.05 | -0.013 | 0.25 | -0.0095 | 0.044 |
| SEM | 8.5e-4 | 0.0017 | 0.009 | 0.011 | 0.0007 | 0.001 | 0.0007 | 0.04 | 0.034 | 0.008 | 0.0011 | 0.0027 | 0.045 | 0.0023 | 0.02 |
| SSIM (FMNIST → VGG-16 ImageNet) | 0.57 | 0.46 | 0.5 | 0.87 | 0.64 | 0.67 | 0.64 | 0.5 | 0.38 | 0.8 | 0.36 | 0.64 | 0.66 | 0.12 | 0.2 |
| SEM | 0.012 | 0.011 | 0.015 | 0.0056 | 0.01 | 0.015 | 0.011 | 0.015 | 0.03 | 0.009 | 0.01 | 0.02 | 0.018 | 0.0049 | 0.024 |
| RK (FMNIST → VGG-16 ImageNet) | -0.0023 | -0.0098 | -0.0097 | 0.028 | -0.0025 | -0.0017 | -0.0025 | 0.005 | -0.045 | 0.25 | 0.03 | 0.0045 | 0.32 | 0.066 | 0.14 |
| SEM | 0.0017 | 0.0025 | 0.02 | 0.018 | 0.0023 | 0.0016 | 0.002 | 0.033 | 0.024 | 0.004 | 0.0035 | 0.0018 | 0.034 | 0.0053 | 0.019 |

Table 3: **Test-time Explanation Similarity Metrics.** We observe visual similarity but no ranking similarity. We show each metric along with the standard error of the mean calculated for 190 examples. FMNIST → MNIST model means a comparison of FMNIST attributions for an FMNIST model with FMNIST attributions derived from *an MNIST model*. We present both SSIM and Rank correlation metrics.

**Bug Implementation.** We consider 4 dataset-model pairs: a BVD-CNN trained on MNIST, Fashion MNIST, the Birds-vs-dogs data, and lastly a VGG-16 model trained on ImageNet. We present results on Fashion MNIST. Concretely, we compare 1) feature attributions of Fashion MNIST examples derived from a model trained on Fashion MNIST, and 2) feature attributions of Fashion MNIST examples for models trained on MNIST, the birds-vs-dogs dataset, and ImageNet.

**Results.** As shown in Figure 8, we observe visual similarity between in-domain Fashion MNIST attributions, and attributions for these samples on other models. As seen in Table 3, we observe visual similarity, particularly for the VGG-16 model on ImageNet, but essentially no correlation in feature ranking.

**Insights from Human Subject Study: users use prediction labels, not the attributions.** For the domain shift study, we show participants attribution of dogs that were not used during training, and

whose breeds differed from those that the model was trained to predict. We find that users do not recommend a model under this setting due to wrong prediction labels (Figure 5).

## 6   Discussion & Conclusion

Debugging machine learning models remains a challenging endeavor, and model explanations could be a useful tool in that quest. Even though a practitioner or a researcher may have a large class of explanation methods available, it is still unclear which methods are useful for what bug type. This work aims to address this gap by first, categorizing model bugs into: data, model, and test-time contamination bugs, then testing feature attribution methods, a popular explanation approach for DNNs trained on image data, against each bug type. Overall, we find that feature attribution methods are able to diagnose the spatial spurious correlation bug tested, but do not conclusively help to distinguish mislabelled examples for normal ones. In the case of model contamination, we find that certain feature attributions that perform positive aggregation while computing feature relevance with modified back-propagation produce attributions that are invariant to the parameters of the higher layers for a deep model. This suggests that these approaches might not be effective for diagnosing model contamination bugs. We also find that attributions of out-of-domain inputs are similar to attributions for these inputs on an in-domain model, which suggests caution when visually inspecting these explanations, especially for image tasks. We also conduct human subject tests to assess how well end-users can use attributions to assess model reliability. Here we find that the end-users relied, primarily, on model predictions for diagnosing model bugs.

Our findings come with certain limitations and caveats. The bug characterization presented only covers the standard supervised learning pipeline and might not neatly capture bugs that result from a combination of factors. We only focused on feature attributions: however, other methods such as approaches based on 'concept' activation [28], model representation dissection [9], and training point ranking [32, 48, 69] might be more suited to the debugging tasks studied here. Indeed, initial exploration of the 'concept' activation method TCAV and training point ranking based on influence functions suggests that these approaches are promising (See Appendix for analysis). For the human subject experiments, our finding that the participants mostly relied on the labels instead of the feature attributions might be a consequence of the dog breed classification task. It is unclear whether participants would still rely of model predictions for tasks in which they have no expertise or prior knowledge.

The goal of this work is to provide guidance for researchers and practitioners seeking to use feature attributions for model debugging. We hope our findings can serve as a first step towards more rigorous approaches for assessing the utility of explanation methods.

## Broader Impact

Predictive models are increasingly being investigated, sometimes legally regulated for deployment in critical settings. Interpretability methods promise to provide insights about how models make decisions. This may increase user trust and provide the evidence needed to ensure that models deployed in mission-critical settings function adequately. The goal of our work is to investigate this literature with a critical eye: can attribution methods signal that there may be issues with the model, data or at test-time setting? We provide both quantitative and qualitative approaches to evaluate many popular attribution methods in order to provide practitioners and researchers with a set of debugging tests which may be used in validation. We hope our work is one of the first of many to bridge the gap between methods developed in academia and practical usage of those methods in the real world.

## Acknowledgments and Disclosure of Funding

We thank Hal Abelson, Danny Weitzner, Taylor Reynolds, and Anonymous reviewers for feedback on this work. We are grateful to the MIT Quest for Intelligence initiative for providing cloud computing credits for this work. JA is supported by the Open Philanthropy Fellowship.

## Footnotes

[1]We encourage readers to consult the more complete manuscript on the arXiv.

[2]We refer to: https://github.com/adebayoj/explaindebug.git, for code to replicate our findings and experiments.

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
