[Supplementary Material]

# Appendix

## Table of Contents

## A   Additional Discussion of Bug Categorization Setup

Figure 9: We show the debugging schematic here.

In this section, we provide two additional formalization of bugs beyond those discussed in the main document: Frozen layers and Pre-Processing Mismatch. In the case of frozen layers, this is a bug whereby one of the layers of a deep network is accidentally kept fixed during training. In the case of pre-processing mismatch, a test-input can be pre-processed under settings that are different from those used for the training data. In such a case, it is possible the the model will produce wrong predictions due to the mismatch in input pre-processing. We formalize both these bugs in Table 4.

The examples presented in Table 4 are specific instantiation of bugs based on the categorization that we present; as expected, several other examples can be proposed under the same categorization.

| Bug Class | Specific Examples | Formalization |
|---|---|---|
| Data Contamination | Spurious Correlation | $\arg\min_{\theta} L(X_{\text{spurious artifact}}, Y_{\text{train}}; \theta)$ |
| Data Contamination | Labelling Errors | $\arg\min_{\theta} L(X_{\text{train}}, Y_{\text{wrong label}}; \theta)$ |
| Model Contamination | Initialized Weights | $f_{\theta\text{init}}(x_{\text{test}})$ |
| Model Contamination | Frozen Layers | $\arg\min_{\theta\text{frozen}} L(X_{\text{train}}, Y_{\text{wrong label}}; \theta)$ |
| Test-Time Contamination | Out of Distribution (OOD) | $f_{\theta}(x_{\text{OOD}})$ |
| Test-Time Contamination | Pre-Processing Mismatch (PM) | $f_{\theta}(x_{\text{PM}})$ |

Table 4: **Example bugs for different contamination classes.** We show different examples of bugs under the different contamination classes: data, model and test-time. We also formalize these bugs for the traditional supervised learning setting.

## B   Detailed Overview of Attribution Methods

We now provide a detailed discussion of the explanation methods that we assess in this work. For each method we also provide the hyper-parameter values that we use in each case. First, we re-implemented all methods in a single code base. We then benchmark our implementation with the public open source implementations of these methods. Ultimately, we used the public implementation of the these methods as released by their authors. For gradient based methods and the corresponding LRP variants, we use the innvestigate python package. For LIME we used the publicly available LIME-IMAGE package. For Kernel Shap and Expected Gradients, we used the public SHAP package.

Recall that an attribution functional, $E : \mathcal{F} \times \mathbb{R}^d \times \mathbb{R} \to \mathbb{R}^d$, maps the input, $x_i \in \mathbb{R}^d$, the model, $F$, output, $F_k(x)$, to an attribution map, $M_{x_i} \in \mathbb{R}^d$.

## Overview of Methods

Figure 10: We show five input examples from the birds-vs-dogs dataset along with the attributions for a normally trained model.

**Gradient (Grad) & Variants.** We consider:

- The *Gradient (Grad)* [8, 59] map, $|\nabla_{x_i} F_i(x_i)|$. The gradient map is a key primitive upon which several other methods are derived. Overall, the gradient map quantifies the sensitivity of the output, typically a logit score for DNNs trained for classification, to each dimension of the input.

- *SmoothGrad (SGrad)* [62], $E_{\text{sg}}(x) = \frac{1}{N} \sum_{i=1}^{N} \nabla_{x_i} F_i(x_i + n_i)$ where $n_i$ is sampled according to a random Gaussian noise. we considered 50 noisy inputs, selected the standard deviation of the noise to be $0.15 * $ input range. Here input range refers to the difference between the maximum and minimum value in the input. For the models considered, these inputs are typically normalized to be in the range $[-1, 1]$.

- *SmoothGrad Squared (SGradSQ)* [26], the element-wise square of SmoothGrad: $E_{\text{SGradSQ}}(x) = E_{\text{sg}}(x) \odot E_{\text{sg}}(x)$. We set the parameters of SmoothGrad as discussed in the previous item.

- *VarGrad (VGrad)* [1], the variance analogue of SmoothGrad: $E_{\text{VGrad}}(x) = \mathbb{V}(\tilde{x})$, where $\mathbb{V}$ is the variance operator, $\tilde{x} = \nabla_{x_i} F_i(x_i + n_i)$, and $n_i$ is sampled according to a random Gaussian noise. Here we consider 50 noise input examples, set the noise parameter as a Gaussian with mean zero, and noise scale similar to SmoothGrad: $0.15 *$ input range.

- *Input-Grad* [58] the element-wise product of the gradient and input $|\nabla_{x_i} F_i(x_i)| \odot x_i$. Several other methods approximate and have been shown to be equivalent to this product. For example, for a DNN with all ReLU activations, LRP-Z, a variant of layer-wise relevance propagation that we discuss in the modified back-propagation section is equivalent to Input-Grad [29].

- *Integrated Gradients (IntGrad)* which sums gradients along an interpolation path from the "baseline input", $\bar{x}$, to $x_i$: $M_{\text{IntGrad}}(x_i) = (x_i - \bar{x}) \times \int_0^1 \frac{\partial S(\bar{x} + \alpha(x_i - \bar{x}))}{\partial x_i} d\alpha$. For integrated gradients we set the baseline input to be a vector containing the minimum possible values across all input dimensions. This often corresponds an all-black image. The choice of a baseline for IntGrad is not without controversy; however, we follow this setup since it is one of the more widely used baselines for image data.

- *Expected Gradients (EGrad)* which computes IntGrad but with a baseline input that is an expectation over the training set: $x_i$: $M_{\text{EGrad}}(x_i) = \int_{x'} \left( (x_i - \bar{x}) \times \int_0^1 \frac{\partial S(\bar{x} + \alpha(x_i - \bar{x}))}{\partial x_i} d\alpha \right) p_D(x') dx'$. As we see, this is equivalent to IntGrad, but where the multiple baselines are considered. We use 200 examples from the training set as baseline.

For the methods considered under gradients and variants, we re-implemented all methods in Tensorflow. We also benchmark our implementation with those provided in the open-source "innvestigate" python package. We have included example scripts that compute the attribution maps using the open-source innvestigate package. In the case of expected gradients, we bench-marked our implementation against a public one in the SHAP python package.

**Surrogate Approaches.** We consider:

- LIME [49] locally approximate $F$ around $x_i$ with a simple function, $g$, that is then interpreted. LIME corresponds to: $\arg\min_{g \in G} L(f, g, \text{pert}(x_i)) + \Omega(g)$, where $\text{pert}(x_i)$ local perturbations of the input $x_i$, and $\Omega(g)$ is a regularizer. Overall, recent work has shown that, in the tabular setting, LIME approximates the coefficients of a black-box linear model with high probability. In our empirical implementation we follow the open source lime-image package. Here to account for high dimensions, the input image is first segmented into 50 segments and the local approximation $g$ is fit around input perturbations with 50. We experimented with $5, 10, 15, \& 25$ dimensions as well. Overall, we found the LIME with 50 segments to be more stable for the input data sizes that we consider. We use 1000 samples in model fitting.

- SHAP [37] Similar to LIME, SHAP provides a local approximation around a single input. The local model is then interpreted as a form of explanation. SHAP unifies LIME and several under methods under the same umbrella and turns out to be a tractable approximation to the Shapley Values [56]. We use 1000 samples in model fitting.

**Modified Back-Propagation.** These class of methods apportion the output into 'relevance' scores, for each input dimension, using back-propagation. *DConvNet* [71] & *Guided Back-propagation (GBP)* [63] modify the gradient for a ReLU unit. Alternatively, *Layer-wise relevance propagation (LRP)* methods specify 'relevance' rules that modify the regular back-propagation. For example, let $r_l$ be the unit relevance at a layer $l$, the $\alpha\beta$ rule is: $r_l(x_i) = (\alpha Q_l^+ - \alpha Q_l^-) r_{l+1}(x_i)$, where $Q$ is a 'normalized contribution matrix'; $Q^+$ and $Q^-$ are the matrix $Q$ with only positive and negative entries respectively. We consider *LRP-EPS* since *LRP-Z* (using the $z$ rule) is equivalent to Input-Grad for ReLU networks [31]. *PatternNet (PNet)* and *Pattern Attribution (PAttribution)* decompose the input into signal and noise components, and back-propagate relevance for only the signal component. We now provide additional detail:

- *Deconvnet [71] & Guided Backpropagation (GBP) [63]* both modified the backpropagation process at ReLu units in DNNs. Let, $a = \max(0, b)$, then for a backward pass, $\frac{\partial l}{\partial s} = 1_{s>0} \frac{\partial l}{\partial b}$, where $l$ is a function of $s$. For Deconvnet, $\frac{\partial l}{\partial s} = 1_{\frac{\partial l}{\partial s} > 0} \frac{\partial l}{\partial b}$, and for GBP, $\frac{\partial l}{\partial s} = 1_{s>0} 1_{\frac{\partial l}{\partial s} > 0} \frac{\partial l}{\partial b}$.

- *PatternNet & Pattern Attribution, Kindermans et al. [30].* PatternNet and Pattern Attribution first estimate a 'signal' vector from the input; then, the attribution (in the case of Pattern Attribution) corresponds to the covariance between the estimated signal vector, and the output, $F(x)$, propagated all the way to the input. We use the innvestigate package implementation of PatternNet and Pattern Attribution, which we bench marked against a re-implemented version.

- *DeepTaylor [42].* DeepTaylor describes a family of methods that iteratively compute local Taylor approximations for each output unit in a DNN. These approximation produce unit attribution that are then propagated and redistributed all the way to the input. We use the innvestigate package implementation of DeepTaylor, which we bench marked against a re-implemented version.

- *Layer-wise Relevance Propagation (LRP) & Variants, Alber et al. [3], Bach et al. [6].* LRP attribution methods iteratively estimate the relevance of each unit of a DNN starting from the penultimate layer all the way to the input in a message-passing manner. We consider 4 variants of the LRP method that correspond to different rules for propagating unit relevance. In our detailed treatment in the appendix, we provide definitions and restate previous theorems that show that certain variants of LRP are equivalent to Gradient⊙Input for DNNs with ReLU non-linearities. We use the innvestigate package implementation of LRP-Z, LRP-EPS, $\alpha - \beta$-LRP and a preset-flat variant, which we bench marked against a re-implemented version. We kept the default hyper-parameters that the innvestigate package provides.

**Attribution Comparison.** We measure visual and feature ranking similarity with the structural similarity index (SSIM) and Spearman rank correlation metrics, respectively.

**Visualization Attributions and Normalization.** Here and in the main text we show attributions in a single color scheme: either Gray Scale or a White-Red scheme. We do this to prevent visual clutter. For all the metrics we compute, we normalize attributions to lie between [0, 1] for SSIM and [-1, +1] for attributions that return negative relevance.

# C    Datasets & Models

Here we provide a detailed overview of the data set and models used in our experiments. For detailed overview of model hyper-parameters and details, we refer to the github repository: https://github.com/adebayoj/explaindebug.git. We provide additional details for each model bug in the section where the bug is discussed.

## C.1    Datasets.

**Birds-Vs-Dogs Dataset.**    We consider a birds vs. dogs binary classification task for all the data contamination experiments. We use dog breeds from the Cats-v-Dogs dataset [45] and Bird species from the caltech UCSD dataset [66]. These datasets come with segmentation masks that allows us to manipulate an image. All together, this birds-vs-dogs dataset consists of 10k inputs (5k dog samples and 5k bird samples). We use 4300 data points per class, 8600 in total for training, and split the rest evenly for a validation and test set.

**Modified MNIST and Fashion-MNIST Datasets.**    For the test-time contamination tests, we modify the MNIST and Fashion-MNIST datasets to have 3 channels and derive attributions from this three-channel version of MNIST from a VGG-16 model.

**ImageNet Dataset.**    We use two 200 images from the ImageNet [52] validation set for the model contamination tests.

**User Study Dogs Only Dataset.**    For the user study alone, we restrict our attention to 10 dog classes. We used a modified combination of the Stanford dogs dataset [27] and the Oxford Cats and Dogs datasets [46]. We restrict to a 10-class classification task consisting of the following breeds: *Beagle, Boxer, Chihuahua, Newfoundland, Saint Bernard, Pugs, Pomeranian, Great Pyrenees, Yorkshire Terrier, Wheaten Terrier.* We are able to create spurious correlation bugs by replacing the background in training set images. We consider 10 different background images representing scenes of: *Water Fall, Bamboo Forest, Wheat Field, Snow, Canyon, Empty room, Road or Highway, Blue Sky, Sand Dunes, and Track.*

## C.2    Models.

**BVD-CNN**    For the data contamination tests, we consider a CNN with 5 convolutional layers and 3 fully-connected layers with a ReLU activation functions but sigmoid non-linearity in the final layer. We train this model with an Adam optimizer for 40 epochs to achieve test accuracy of 94-percent. For ease of discussion, we refer to this architecture as *BVD-CNN*. We use a learning rate of 0.001 and the ADAM optimizer. This is the standard BVD-CNN architecture setup that we consider.

**User Study Model.**    Here we use a ResNet-50 model that was fine-tuned on the dogs only dataset to generate all the attributions for the images considered. Please see public repository for model training script.

# D   Data Contamination

## D.1   Data Contamination: Spurious Correlation Artifacts

**Confirming Spurious Model.**   To confirm that the BVD-CNN trained on a spurious data indeed uses the sky and bamboo-forest signals, we tested this model on a test set of only Sky and Bamboo-forest with no dogs or birds images. On this test-set, we obtain a 97 percent spurious accuracy. As expected, we also maintain this 97 accuracy for a test-set that has the birds and dogs inserted on the appropriate backgrounds. Please see the additional figures section for more examples.

## D.2   Data Contamination: Mislabelled Examples

**Bug Implementation.** We train a BVD-CNN model on a birds-vs-dogs dataset where 10 percent of training samples have their labels flipped. The model achieves a 94.2, 91.7, 88 percent accuracy on the training, validation and test sets. We show additional examples in later paper of the supplemental material.

# E   Model Contamination

**Model Contamination Tests.**   The model contamination tests capture defects that occur in the parameters of a model. Here, we consider the simple setting where a model is accidentally reinitialized during or while it is being used. As expected, such a bug will lead to observable accuracy differences. However, the goal of these classes of tests, especially in the model attribution setting, is to ascertain how well a model attribution is able to identify models in different parameter regimes.

# F   Test-Time Contamination

**Test-Time Contamination Tests.**   This class of tests captures defects that occur at test-time. A common test-time bug is one where the test input has been pre-processed in a way that was different than the training data. In other cases, a model trained in one setting receives inputs that are out of domain. We show additional figures for this setting in the later figures section.

# G   Other Methods

**Concept and Influence Functions.   Influence Function and Concept Methods.** We focused on attribution methods to keep our inquiry thematically focused. Here, we test: i) influence functions (IF) [32] for training point ranking, and ii) the concept activation (TCAV) approach [28]. We assess IF under spurious correlation, mislabelled examples, and domain shift. We test TCAV under the spurious correlation and model contamination setting (see Figure 11). TCAV ranking for the background concept indicates that it might be able to identify spurious correlation. We find that IF shows association regarding spurious correlation for background inputs. For example, for a given input with the spurious background, we measure the fraction of the top 1 percent of training points (86) that are spurious. We find, on 50 examples, that 91 percent (2.4 standard error) of the top 1 percent of training points are spurious. Similarly, we use the self-influence metric to assess mislabelling (see arXiv:2002.08484), and find that for 50 examples, we need to check an average of 11 percent of the training set (5.9 standard error). These results suggest that both methods might be effective for model debugging. We caution, however, that significant additional empirical assessment is required to confirm that both methods are effective for the bugs tested.

Figure 11: Assessing TCAV and influence functions.

# H   User-Study Overview

**Task & Setup:** We designed a study to measure people's ability to assess the reliability of classification models using feature attributions. People were asked to act as a quality assurance (QA) tester for a hypothetical company that sells animal classification models. Participants were shown the original image, model predictions, and attribution maps for 4 dog breeds at time. They then rated how likely they are to recommend the model for sale to external customers using a 5 point-Likert scale. They provided a rationale for their decision, and participants chose from 4 pre-created answers (Figure 5-b) or filled in a free form answer. Participants self-reported their ML experience and answered 3 questions aimed at verifying their expertise.

**Methods:** We focus on a representative subset of methods: Gradient, Integrated Gradients, and SmoothGrad. Given the scope of available attribution methods, we use the randomization tests to help narrow down to a selection of methods. Amongst the methods that performed better than others in the randomization tests, we observe two groups: 1) Gradient & Variants (SmoothGrad & VarGrad) and methods that approximate the gradient like LIME and SHAP, and 2) Integrated Gradients, Expected Gradient, LRP-EPS, & LRP-Z that all approximate the Input⊙Gradient. Consequently, we select from amongst these methods to perform the debugging tests. Our selection criteria was as follows: 1) We focus on methods that apply in broad generality and from which other methods are derived (Gradient); 2) We focus on methods that have been previously used for model debugging in past literature (e.g., Integrated Gradients [54, 64]; and finally, 3) We select methods that have been shown to have desirable against manipulation (SmoothGrad) [19, 70]. On the basis of these selection criteria, we pick: *Gradient, SmoothGrad, & Integrated Gradient* as the methods to assess in our proposed debugging tests. These three methods allow us to characterize important classes of method to which each method belongs while providing us the flexibility to apply across a variety of tasks.

Figure 12: **Schematic of** 5 **different model conditions considered in the user study**. We show model attributions across each model condition and for a diverse array of inputs.

**Recruiting Participants.** We recruited participants through the university mailing list of a medium-sized North-American university. In total, 54 participants completed the study.

**Machine Learning Expertise.** We asked participants to self-assess and report their level of expertise in machine learning. In addition, we asked a simple test question on the effect of parameter regularization. More then 80% of the participants reported past experience with machine learning.

**Task, Procedure, & Structure of Experiment.** As originally noted, the task at hand is that of classifying images of Dogs into different breeds. We manually selected 10 breeds of dogs based on authors' familiarity. All model conditions were trained to perform a 10-way classification task. As part of the recruitment, participants were directed to an online survey platform. Once the task was clicked, they were presented a consent form that outlined the aim and motivation of the study. We then provided a quick guide on the breeds of dogs the model was trained on, and the set-up and study interface. We show these training interfaces in Figures 13 and 14.

Participants were asked to take on the role of a quality assurance tester at a machine leaning start-up that sells animal classification models. The goal of the study was then for them to assess the model, using both the model labels and the attributions provided. For each condition, each participant was shown images of dogs along with model labels and attributions for a specific model condition. The participant was then asked: **using the output and explanation of the dog classification model below, do you think this specific model is ready to be sold to customers?** Participants were asked to then respond on a 5-point Likert scale with options ranging from Definitely Not to Definitely. A second sub-question also asked participants to provide the motivation for their choice. Taken together, each participant was asked 21 unique questions and 1 repeat as an attention check.

**Datasets.** We create a modified combination of the Stanford dogs dataset [27] and the Oxford Cats and Dogs datasets [46] as our primary data source. We restrict to a 10-class classification task consisting of the following breeds: *Beagle, Boxer, Chihuahua, Newfoundland, Saint Bernard, Pugs, Pomeranian, Great Pyrenees, Yorkshire Terrier, Wheaten Terrier.* In total, each class of dogs consists of 400 images in total making 4000 images. We had the following train-validation-test split: (350, 25, 25) images. Each split was created in an IID manner. The Oxford portion of the Dogs datasets includes a segmentation map that we used to manipulate the background of the images. The Stanford dogs dataset includes bounding box coordinates that we used to crop the images for the spurious versions of the data sets that we created. We now overview the data condition for each model manipulation (bug) that we consider:

This is the question you will answer throughout this task.

This is the prediction of the ML on the image.

Using the output and explanation of the dog classification model below, do you think this specific model is ready to be sold to customers?

Algorithm Prediction: Beagle

Algorithm Explanation

Algorithm Prediction: Boxer

Algorithm Explanation

This kind of explanation is called a heat map. It shows the parts of the image that the ML relied on to make the prediction.

Pick out of these choices.

DEFINITELY NOT | PROBABLY NOT | UNSURE/MAYBE | PROBABLY | DEFINITELY

What were your motivation for your response above?

○ On some or all of the images, the dog breed was wrong.
○ The dog breeds were correct.
○ The explanation did not highlight the part of the image that I expected it to focus on.
○ Other, please specify

Say why you made the choice above.

Figure 13: **Training Interface for the User-Study**.

Algorithm Prediction: Beagle

Algorithm Explanation

For the Blue/Red images, the red dots are positive evidence, and the blue dots are negative evidence.

In this example, the red dots are evidence for why this image is a Beagle. The blue dots are evidence for a breed other than Beagle.

Algorithm Prediction: Beagle

Algorithm Explanation

The white portions are important parts that the ML relied on.

Figure 14: **Training Interface for the User-Study**.

- Normal Model: In this case, we have the standard dataset without alteration. This is the control.
- Top-Layer: Here we make no changes to the data set. The bug corresponds to a model parameter bug.
- Random Labels: Here we created a version of the dataset where all the training labels were randomized.

Figure 15: **User Study Demographics.** We show counts and breakdown of the demographics of the participants in the User Study.

Figure 16: **User Study Demographics.** We show counts and breakdown of the demographics of the participants in the User Study.

- Spurious: Here we replace the background on all training points with the background that was pre-associated with that specific class. Note, here that we also test on the spurious images as well.

- Out-of Distribution. Here we apply a normal model on breeds of dogs that were not seen during training.

**Bugs:** We tested the following bugs:

- **Control Condition**: Normal Model (ResNet-50 trained on normal data).

- **Model Contamination Test 1**: Top Layer Random (ResNet-50 with reinitialized last layer).

- **Data Contamination Test 1**: Random Labels (ResNet-50 trained on data with randomized labels).

## Machine Learning Experience

## Attribution Familiarity

Figure 17: **Machine Learning Experience and Familiarity with Feature Attributions.** We show counts and breakdown of the demographics of the participants in the User Study.

- **Data Contamination Test 2** Spurious (ResNet-50 trained on data where all training samples have spurious background signal).
- **Test-Time Contamination Test**: Normal model tested on attributions from out of distribution (OOD) dog breeds.

For the spurious correlation setting, we consider each breed and an associated background as shown in Table H.

| Dog Species | Associated Background |
|---|---|
| Beagle | Canyon |
| Boxer | Empyt Room |
| Chihuahua | Blue Sky |
| Newfoundland | Sand Dunes |
| Saint Bernard | Water Fall |
| Pugs | High Way |
| Pomeranian | Track |
| Great Pyrenees | Snow |
| Yorkshire Terrier | Bamboo |
| Wheaten Terrier | Wheat Field |

**Data Analysis.** For each model manipulation, we performed a one-way Anova test and a post-hoc Tukey Kramer test to assess the effect of the attribution on the ability of participants to reject a defective model. There was a statistically significant difference between attribution maps as determined by one-way ANOVA computed per each manipulation. Within the *normal*, there was a statically significance difference between attribution maps (one-way ANOVA ($F(2, 54) = 14.35$, $p < 0.0001$)). A Tukey post hoc test revealed that participants reported being more likely to recommend Gradient ($\mu = 3.88$, $p < 0.0001$) and SmoothGrad ($\mu = 3.77$, $p < 0.0001$) attribution maps over Integrated-Gradients ($\mu = 2.805$). There was no statistically significant difference between Gradient and SmoothGrad ($p = 0.815$). Within the *out-Distribution*, there was not a statically significance difference between attribution maps ($p = 0.109$). Within the *Random-Labels*, there was a statically significance difference between attribution maps (one-way ANOVA ($F(2, 54) = 7.66$, $p < 0.0007$)). A Tukey post hoc test revealed that participants reported being more likely to recommend Integrated-Gradients ($\mu = 1.94$, $p < 0.0001$) over SmoothGrad ($\mu = 1.31$). There was no statistically significant difference between Gradient and SmoothGrad ($p = 0.199$) and between Integrated-Gradient and Gradient ($p = 0.077$). Within the *Spurious*,

there was a statically significance difference between Attribution maps (one-way ANOVA ($F(2, 54) = 15.9$, $p < 0.0001$)). A Tukey post hoc test revealed that participants reported being more likely to recommend Integrated-Gradients ($\mu = 3.05$, $p < 0.0001$) over SmoothGrad ($\mu = 1.98$) and Integrated-Gradient ($\mu = 3.05$, $p = 0.0011$) over Gradient ($\mu = 2.35$,). There was no statistically significant difference between Gradient and SmoothGrad ($p = 0.1370$). Within the *Top-Layer* manipulation, there was not a statically significance difference between attribution maps ($p = 0.085$).

**User Study: Figures for Different Model Conditions**   Along with the collection of additional figures in the next section, we show figures for different model conditions and attribution combinations. The images shown were the ones used in the study that we performed.

# I   Collection of Additional Figures

Figure 18: **Spurious Correlation Bug.** Spurious Image and Spurious Background Only for 4 image setups under the Gray Scale visualization.

**Spurious Correlation**

Figure 19: **Spurious Correlation Bug.** Spurious Image and Spurious Background Only for 4 image setups under the White-Red visualization..

**Spurious Correlation**

Figure 20: **Spurious Correlation Bug.** Spurious Image and Spurious Background Only for 4 image setups under the Gray Scale visualization.

Figure 21: **Mislabelled Examples Bug.** Figure shows the input in the top row, along with attribution visualization for this input under two visualization schemes. We show both Gray Scale and the White-Red visualization scheme here.

Figure 22: **Mislabelled Examples Bug.** Figure shows the input in the top row, along with attribution visualization for this input under two visualization schemes.

Figure 23: **Mislabelled Examples Bug.** Figure shows the input in the top row, along with attribution visualization for this input under two visualization schemes.

Figure 24: **Mislabelled Examples Bug.** Figure shows the input in the top row, along with attribution visualization for this input under two visualization schemes.

Figure 25: **Mislabelled Examples Bug.** Figure shows the input in the top row, along with attribution visualization for this input under two visualization schemes.

Figure 26: **Additional Model Contamination Quantitative Metrics.** Similarity metrics computed for 200 images across 17 attribution types for the VGG-16 model trained on ImageNet. We use SSIM to quantify visual similarity. We also show the rank correlation metric with and without absolute values. We see that for certain methods, like those that modify backprop, across these series of metrics, these methods show high similarity. We also show the normalized norm difference for each method computed as: $\frac{||e_{orig} - e_{rand}||}{||e_{orig}||}$. This is the normalized difference in $2-$norm between the original attribution and the attribution derived from a (partially) randomized model. Note that we do not include Expected Gradients for VGG-16 experiments because it was too computationally intensive.

Figure 27: An example Junco Bird Image.

Figure 28: **Model Contamination Bug VGG-16 on ImageNet.** Cascading parameter randomization on the VGG-16 model for a Junco bird example (We use the Gray Scale visualization scheme here).

Figure 29: **Model Contamination Bug VGG-16 on ImageNet.** Cascading parameter randomization on the VGG-16 model for a Junco bird example (We use the Red visualization scheme here)

Figure 30: An example Bug Image.

Figure 31: **Model Contamination Bug VGG-16 on ImageNet.** Cascading parameter randomization on the VGG-16 model for a bug example.

Figure 32: **Model Contamination Bug VGG-16 on ImageNet.** Cascading parameter randomization on the VGG-16 model for a bug example.

Figure 33: An example Dog-Cat Image.

Figure 34: **Model Contamination Bug VGG-16 on ImageNet.** Cascading parameter randomization on the VGG-16 model for the dog-cat example.

Figure 35: **Model Contamination Bug VGG-16 on ImageNet.** Cascading parameter randomization on the VGG-16 model for the dog-cat example.

Figure 36: An example Mouse-Cat Image.

Figure 37: **Model Contamination Bug VGG-16 on ImageNet.** Cascading parameter randomization on the VGG-16 model for the Mouse-Cat example.

Figure 38: **Model Contamination Bug VGG-16 on ImageNet.** Cascading parameter randomization on the VGG-16 model for the Mouse-Cat example.

Figure 39: An example Corn Image.

Figure 40: **Model Contamination Bug VGG-16 on ImageNet.** Cascading parameter randomization on the VGG-16 model for the Corn example.

Figure 41: **Model Contamination Bug VGG-16 on ImageNet.** Cascading parameter randomization on the VGG-16 model for the Corn example.

Figure 42: An example Dog biting on a Bucket.

Figure 43: **Model Contamination Bug VGG-16 on ImageNet.** Cascading parameter randomization on the VGG-16 model for the Dog example.

Figure 44: **Test-time Randomization Example for Fashion MNIST.** We show Fashion MNIST attributions for 4 different models: A fashion MNIST model (trained on fashion MNIST), an MNIST Model (trained on MNIST), a birds-vs-dogs model (trained on birds-vs-dogs dataset), and a VGG-16 model trained on IMAGENET.

Figure 45: **Test-time Randomization Example for MNIST.** We show MNIST attributions for 4 different models: A fashion MNIST model (trained on fashion MNIST), an MNIST Model (trained on MNIST), a birds-vs-dogs model (trained on birds-vs-dogs dataset), and a VGG-16 model trained on IMAGENET.

Figure 46: **Test-time Randomization Example for Fashion MNIST.** We show Fashion MNIST attributions for 4 different models: A fashion MNIST model (trained on fashion MNIST), an MNIST Model (trained on MNIST), a birds-vs-dogs model (trained on birds-vs-dogs dataset), and a VGG-16 model trained on IMAGENET.

**Input**

Figure 47: **Test-time Randomization Example for Fashion MNIST.** We show Fashion MNIST attributions for 4 different models: A fashion MNIST model (trained on fashion MNIST), an MNIST Model (trained on MNIST), a birds-vs-dogs model (trained on birds-vs-dogs dataset), and a VGG-16 model trained on IMAGENET.

**Input**

Figure 48: **Test-time Randomization Example for Fashion MNIST.** We show Fashion MNIST attributions for 4 different models: A fashion MNIST model (trained on fashion MNIST), an MNIST Model (trained on MNIST), a birds-vs-dogs model (trained on birds-vs-dogs dataset), and a VGG-16 model trained on IMAGENET.

Figure 49: **Test-time Randomization Example for Fashion MNIST.** We show Fashion MNIST attributions for 4 different models: A fashion MNIST model (trained on fashion MNIST), an MNIST Model (trained on MNIST), a birds-vs-dogs model (trained on birds-vs-dogs dataset), and a VGG-16 model trained on IMAGENET.

**Normal Model: Gradient**

Figure 50: Images used as part of the user study.

**Normal Model: SmoothGrad**

Figure 51: Images used as part of the user study.

**Normal Model: Integrated Gradients**

Figure 52: Images used as part of the user study.

**Top Layer Random : Gradient**

Figure 53: Images used as part of the user study.

**Top Layer Model Random: SmoothGrad**

Figure 54: Images used as part of the user study.

**Top Layer Random: Integrated Gradients**

Figure 55: Images used as part of the user study.

**Random Labels : Gradient**

Figure 56: Images used as part of the user study.

**Random Labels: SmoothGrad**

Figure 57: Images used as part of the user study.

**Random Labels: Integrated Gradients**

Figure 58: Images used as part of the user study.

**Spurious : Gradient**

Figure 59: Images used as part of the user study.

**Spurious: SmoothGrad**

Figure 60: Images used as part of the user study.

## Spurious: Integrated Gradients

Figure 61: Images used as part of the user study.

**Out of Distribution : Gradient**

Figure 62: Images used as part of the user study.

**Out of Distribution: SmoothGrad**

Figure 63: Images used as part of the user study.

**Out of Distribution: Integrated Gradients**

Figure 64: Images used as part of the user study.