[Reviews · NeurIPS 2020]

Review 1

Summary and Contributions: The main research question of this paper is that "Can attribution methods reveal bugs in each part of the supervised learning setting?". The paper first categorizes bugs into three classes: data, model, and test-time contamination. Then it conducts both empirical and human experiments on 15 attribution methods and four bug conditions to see how effective the methods can signal the bug issues. The paper then presents some insights from the experiments. For example, most of the tested methods can detect spurious correlation in the training data but cannot detect the problem of mislabeled training examples. Some attribution methods perform well at identifying bugs according to quantitative metrics (visual similarity and rank correlations); however, human participants rely mainly on the wrong predictions rather than the unexpected explanations when they assess the models.

Strengths: - The main research question is clear and worth studying. Together with other existing work, it is driving the community to the next step, from research to applications of explanation methods. - There are both empirical analyses and human experiments in this paper. This is a good quality of papers studying explanation methods. - The paper is well-written, the figures are easy to understand, and the appendix contains sufficient details about the settings and the experimental results. - Many attribution methods were tested using several image datasets.

Weaknesses: The explanation methods studied in this paper are limited to only attribution methods. This is not bad; however, some types of bugs might be more relevant to other explanation classes. For example, Influence functions (explaining by examples) have been found effective to detect domain mismatch (i.e., OOD) and fix mislabeled examples. So, it would be better if the paper expanded the scope to other classes of explanation methods.

Correctness: I have not seen any major problems with the experiments. One additional comment is that, if I'm not mistaken, human participants are allowed to select only one option as a reason for their recommendation in Figure 5b. There could be cases where humans use both labels and explanations as their reasons, but they were allowed to pick only one. Generally, labels are of course more obvious and objective than explanations. That's why labels were selected as reasons in many bug conditions in Fig 5b. If we want to check the real usefulness of explanations, we may need to allow the participants to select more than one options. Alternatively, we may ask only questions where the predicted class is correct (so we can eliminate the wrong label factor).

Clarity: Yes. The paper is well-written and easy to follow.

Relation to Prior Work: Yes it is, as far as I'm concerned.

Reproducibility: Yes

Additional Feedback: Questions: - How can we draw any conclusion from visual similarity and rank correlation scores? For example, at line 212-215, there is a fair visual similarity but no correlation in feature ranking. What is the interpretation of this situation? - When the model prediction was incorrect, which class did the explanation in your experiments explain? The correct class or the predicted class? Also, regarding test-time contamination, how can you decide the predicted class from the models that were trained on very dissimilar datasets of which the classes cannot be mapped to the target classes? Typos: - Line 26: diagnos --> diagnosis / diagnosing? - Line 189: resulkts --> results After reading the author response: Thank you for your answers and clarifications. I think my major concern about generalizability has been addressed. So, overall, I would like to keep my overall score the same (at 7) and I expect that the authors will add the details in the author response to the final manuscript (either in the main paper or in the appendix). Regarding the (only correct predictions) point, I believe that we can also debug the model based on predictions which are correct for wrong reasons. The following paper is a related work: Ross, A. S., Hughes, M. C., & Doshi-Velez, F. (2017). Right for the right reasons: Training differentiable models by constraining their explanations. arXiv preprint arXiv:1703.03717.


Review 2

Summary and Contributions: Many previous work on model interpretability cite “model debugging” as a main motivation, implying that a good interpretation can help us pinpoint and fix issues in a model. However, with a few exceptions, most interpretability papers did not evaluate their methods on this use case, and settled for nice looking visualizations. This paper tests whether existing feature attribution methods help humans assess the reliability of a model, in the face of spurious correlation, mislabelled training examples, model contamination (re-initializing part of the model), and test-time contamination (domain shift). The user study results are largely negative: users often rely on model prediction to make the judgement, not the attributions.

Strengths: Looks at an important question. Very well written: clearly motivated and well structured. Very easy to follow.

Weaknesses: Although I think the paper looked into an important question, I feel like the negative results from the user study largely confirm known issues of the attribution methods and previous results on evaluating interpretation methods. For example, the observation that in a cooperative setting, humans largely rely on model prediction while ignoring explanations is described in many HCI papers including but not limited to “On human predictions with explanations and predictions of machine learning models: A case study on deception detection” by Lai & Tan (FAT* 2019). Many of the empirical assessments are also done in previous papers. I’m having a hard time figuring out what new value this paper provides. The authors consider the bug categorization one of the contributions. But I find this categorization neither sound nor complete. For completeness, this categorization doesn’t consider the possible bugs from more complicated model training paradigms. For example, with unsupervised pretraining, a new type domain shift bug can be introduced by having a bad collection of unsupervised data. If the unsupervised pretraining uses contrastive learning, then a bad choice of negative examples can introduce a new type of bugs. For soundness, it’s not always possible to apply this bug categorization. For example, it is not yet clear if adversarial examples should be attributed to the model or the data. Multiple factors contribute to the failure of feature attribution methods in human reliability assessment. Some factors that are not discussed thoroughly in the paper include: users knowledge about machine learning in general (a person who knows machine learning would be cautious spurious correlation), background information that is provided to the user (does the user know about the training data distribution), and how the explanations are explained to the users (number of examples shown etc). The user study conducted by this paper is not quite thorough.

Correctness: The paper did not propose new methods. Claims seem correct.

Clarity: Yes it's very well written.

Relation to Prior Work: It's missing the discussion of some user studies for interpretation evaluation. One of the key insights in this paper "user rely on prediction not attribution" has been discussed in papers such as Lai & Tan FAT* 2019.

Reproducibility: Yes

Additional Feedback:


Review 3

Summary and Contributions: This paper runs a set of tests on image classification tasks, both computational, qualitative, and user study based. These tests are designed to uncover how well feature attribution methods can be used for debugging. It seems like the primary result from this paper is that these methods can be used for debugging dependence on spurious correlations, but are less effective at catching model or data corruptions.

Strengths: I really like the fact that the authors included so many different attribution methods in their analysis. I also really like that they have a user study component. The ability to see the results of ~15 methods side by side on specific examples is very helpful. The ability to see how users chose "unexpected explanation" for the spurious correlation debugging scenario was also interesting.

Weaknesses: I have several concerns about the paper: 1. It is not cleanly structured. I found myself stumbling through it from one result to the next, not really understanding the layout of the paper until I was done with it. Perhaps grouping all the results by experiment would help? Consider how to minimize the fragmentation of the content. 2. The paper claims to evaluate explanation methods in general, but it really is an evaluation of their use in image classification. This should be made clear. 3. Lastly, it is important to keep in mind that the explanation tools should not always be applied in the same way for all the different tasks that are presented. For example, when trying to identify spurious correlation then what you did makes sense. But for identifying why samples are misclassified you should explain the model's loss, and even then you still need semantic input features to explain, which are much easier to come by in tabular datasets. So I don't think you can make a general claim that XAI attributions are not as useful for finding domain shift based on the results you have, since images are a particularly hard scenario, and you are not explaining the loss. This is just one example, but my broader concern is that more careful thought on how to best apply these methods in each setting could make a big difference on the results.

Correctness: The paper seemed accurate from what I could tell.

Clarity: The paper is not clear in my opinion. See my comments above.

Relation to Prior Work: Reasonably clear.

Reproducibility: Yes

Additional Feedback: Post feedback summary: I appreciated your willingness to clarify the scope of this work (image classification), and discuss the relation to methods that attribute credit to features other than raw input pixels.


Review 4

Summary and Contributions: - The paper investigates if model explanations are effective for users to debug different types of errors - Various explanation methods are compared - A human subject study is performed

Strengths: - The paper addresses very relevant questions on a topic with not so much prior work - The experimental evaluation is extensive and well-designed - Insights of the paper are interesting

Weaknesses: - Some of the evaluation procedures and measures used (e.g. SSIM) seem ad-hoc. Each of the type of bug could and should be investigated more carefully and thoroughly, potentially in a separate paper. - Purely empirical study, lack of deeper technical insights.

Correctness: Claims and methods seem to be correct.

Clarity: The paper is clearly written. However, in my opinion the structure of the paper is suboptimal I understand that many subheadings and boldly marked paragraphs may help to find the information on the first glance, but this structure reminds me of a technical report rather than a well-polished scientific paper.

Relation to Prior Work: The relation to prior work is good.

Reproducibility: Yes

Additional Feedback: ******************************************* Post rebuttal comments: My evaluation of the paper remains unchanged. ****************************************** The paper addresses important practical problems in XAI. The experimental evaluation is very nice and involves humans. However, the interpretation of the results (and theoretical embedding of the observed effects) and the overall technical contribution of the paper are weak.

[Author Response · NeurIPS 2020]

We thank all reviewers for their thoughtful feedback, and for recognizing that the work addresses an important question
(**R2**, **R3**, & **R5**), is well-written (**R2** & **R3**), and considers a breadth of attribution methods (**R2** , **R4**, & **R5**). Below we
address concerns raised as part of the reviews. We believe your insights and comments will make the paper stronger.

**Additional Clarification & Insights (R3 & R5).** Several interpretation approaches have been proposed (see
arXiv:2003.07631 for 35 methods); however, it is unclear when, and for what debugging tasks, an interpretation
method is effective. Our empirical assessment is a step towards addressing this issue, and provides initial evidence
that future work, theoretical and empirical, can build upon. For example, surprisingly, we find that methods that
modify back-propagation to compute relevance are able to detect the spurious background bug despite being essentially
invariant to higher layer parameters. We will clarify in the paper, the contributions that differentiate our work, and
provide concrete recommendations for practitioners. We will highlight where future work can build upon these findings.

**Why only attributions? Influence Function and Concept Methods (R2 &**
**R4).** We focused on attribution methods to keep our inquiry thematically focused.
However, we have tested: i) influence functions (IF) (Koh et. al. 2017) for training
point ranking, and ii) the concept activation (TCAV) approach (Kim et. al. 2018).
We assess IF under spurious correlation, mislabelled examples, and domain shift.
We test TCAV under the spurious correlation and model contamination setting
(see Figure 1). TCAV identifies reliance on the spurious background concept and
shows sensitivity to parameter weights under the model contamination setting.

We find that IF shows association regarding spurious correlation. For example,
for a given input with the spurious background, we measure the fraction of the
top 1 percent of training points (86) that are spurious. We find, on 50 examples,
that 91 percent (2.4 standard error) of the top 1 percent of training points are
spurious. Similarly, we use the self-influence metric to assess mislabelling (see
arXiv:2002.08484), and find that for 50 examples, we need to check an average
of 11 percent of the training set (5.9 standard error). These results suggest that
both methods might be effective for model debugging.

Figure 1: Assessing TCAV and influence functions.

**Structure of the Paper (R4 & R5).** We will streamline the exposition to minimize fragmentation and reduce excessive
reliance on subheading & bolded paragraphs. For example, the discussion of the user study results can be summarized
in a single section instead of across sections 2-5 as it currently stands.

@**R2**: *(Fig 5b):* Users were asked to chose one option, however a free form answer was provided when the need to
select both or other motivations occurred. Participants selected the free form option in 51 out of 1130 responses (4.5%).
Among these responses only 6 reported that a combination of both label or explanation was used to make their choice.
The remainder contained other comments focusing on surprising and unexpected results associated with either the
explanation or the label. *(Only Correct Predictions):* Since the focus of our work is model debugging, it is necessary to
test settings where the predicted class is incorrect. *(Visual Similarity vs. Feature Ranking (212-215).* These findings
indicate that similar portions of the input is important under the two different settings, but that the ranking of the
importance of these dimensions is different. *(Class Prediction).* We always explain the predicted class even when the
predicted class is incorrect. *(Test-Time Debugging).* We explain the 'erroneous' prediction on the out-of-domain model.
Eg., the fashion MNIST input in Fig. 8 is classified as an '8' for an MNIST model, so we explain this target on the
fashion MNIST input for the MNIST model. We will clarify these details in the updated manuscript.

@**R3**: *(Lai & Tan FAccT 2019).* Thanks for the pointer, we will include and discuss the paper. A key difference is that
under our setting, the end user's *sole* task is to assess the model's reliability; however, Lai & Tan consider a setting
where the human and model combine to perform a task. *(Bug Categorization):* We agree that the bug categorization
addresses a simple supervised learning setting. Adversarial examples, and **R3**'s example on unsupervised pre-training
do not fit, cleanly, under the current categorization. We will clarify and state the setting more clearly in the manuscript.
*(User Study Details):* 80% of the participants had either trained a model, taken an ML class, or are ML researchers
(Fig.50 Appendix). We also provided training on model attributions, training data, and the task. We will include this
discussion in the main body of the paper rather than the appendix.

@**R4**: *(Image classification):* We will make it clear that we focus on image classification, and that our results might not
translate to tabular settings with semantically meaningful features. *(Loss):* We performed all mislabeled experiments
again, but this time explain the *loss*. For methods that modify back-propagation, our findings do not change (SSIM
& Rank Correlation $\geq 0.85$). For the other methods, we observe a slight drop: SSIM 0.72-0.81, and rank correlation
0.69-0.83. We agree that this work opens up discussion about other ways of using attribution methods to diagnose bugs.

@**R5**: *(SSIM):* Our use of SSIM to measure visual similarity follows previous work (Adebayo et. al. 2018 & Sixt et. al.
2020) that used it for a similar purpose.

[Meta-Review · NeurIPS 2020]

The paper proposes a number of interesting ideas, and includes really strong empirical results (and user study). However, the reviewers agreed that it could benefit considerably with more precise characterization and theoretical analysis. Please incorporate the feedback from the reviewers in the final version.